# Economic Uncertainty and Firms' Capital Structure: Evidence from China

Chenglin Gao [1] and Takuji W. Tsusaka [2,*]

[1] School of Management, Asian Institute of Technology, Khlong Luang, Pathum Thani 12120, Thailand
[2] School of Environment, Resources and Development, Asian Institute of Technology, Khlong Luang, Pathum Thani 12120, Thailand
* Correspondence: takuji@ait.ac.th or takuji.tsusaka@gmail.com

**Abstract:** This article assesses the effects of economic uncertainty on the corporate capital structure of Chinese-listed firms using a panel dataset of 1138 firms with A-shares traded on the Shanghai Stock Exchange and Shenzhen Stock Exchange for the period 2006–2020 and fixed-effect regression analysis. Economic uncertainty had a negative influence on Chinese firms' debt ratios, especially for non-state-owned enterprises. Furthermore, firms' leverage decreased on average during the 2008 Great Recession, whereas it increased during the 2018–2019 US–China Trade War and the 2020 COVID-19 pandemic. The findings provide quantitative evidence of the effects of economic uncertainty on the capital structure of firms in a transition economy.

**Keywords:** capital structure; debt level; economic uncertainty; panel regression; state-owned enterprise

## 1. Introduction

The Asia Pacific region is a vital component of the world economy, while finance literature has largely focused on Western financial markets. One of the research gaps is the adjustment of corporate capital structure in China under global economic shocks (He and Kyaw 2018). The economic environment in China has changed drastically in the last three decades after the transition of the economic system from a planned economy to a market economy (Zhang et al. 2015). This transition might lead to a different response to economic uncertainty than in developed economies since international investors may participate relatively easily in the financial markets of developed economies (Tran et al. 2018).

Several sources of uncertainty, including economies, politics, government policy, pandemics, and geopolitical conflicts, impact both formal and informal economic environments (Sniazhko 2019; Bloom 2009). Some factors can amplify the magnitude of uncertainty and have a global impact. For instance, the bankruptcy of Lehman Brothers was a milestone event of the Great Recession that had massive adverse effects on the worldwide financial–economic environment. The Great Recession rapidly spread through international financial markets and created significant economic uncertainty (Imbs 2010; Fratzscher and Chudik 2011). Central banks have played crucial roles during financial crises in history and on a global scale to alleviate the vicious cycle of market distress, liquidity freezes, and reductions in the real economy (Reinhart and Rogoff 2009; Aizenman et al. 2016).

Uncertainty in the operating environment directly affects corporate strategies and investment decisions (Rodriguez Lopez et al. 2017; Caldara et al. 2020) including adjustments of corporate capital structure. When economic uncertainty grows, the information gap between borrowers and creditors widens, while firms' future cashflows become more variable, implying a larger chance of default. The increased volatility of expected returns exposes shareholders to higher risks, especially in those countries with higher bankruptcy and monitoring costs (Chen and Chiang 2020), which in turn can lead to rising costs of debt. Firms' investment decisions are also affected by economic uncertainty. In response

to rising uncertainty, firms tend to downscale or delay their investments and expansion plans until the economy stabilizes and becomes predictable (Stokey 2016). This reduction in investments lowers demand for external finance and thus borrowing, affecting the capital structure of the firms.

Since the 2008–2009 Great Recession, there have been two major occasions of global economic uncertainty, namely the US–China Trade War and the spread of COVID-19. Recent studies examined the effects of COVID-19 and found that this crisis had a less impact on businesses that had greater financial flexibility, more cash on hand, and lower debt (Fahlenbrach et al. 2021; Ding et al. 2021). Pettenuzzo et al. (2021) studied how firms coped with the impact of COVID-19 by adjusting financial policies and emphasized that following the advent of the pandemic, US companies discontinued their dividend and share buyback plans in extraordinary numbers.

In emerging markets, previous studies focused on examining the role of target capital structures in alleviating the shocks caused by economic policy uncertainty and how that uncertainty affected corporate investments and the cost of capital. For instance, Zhang et al. (2015) analyzed the effect of EPU (economic policy uncertainty) on Chinese firms from 2003 to 2013 on both the supply and demand sides. Zhang et al. (2019) compared the effects of EPU on financial markets in the USA and China. Kyissima et al. (2019) explored the stability of the capital structure of listed firms in China and investigated how various components of corporate capital structure might explain the variance in leverage ratios among Chinese enterprises.

This emerging context would warrant further research and evidence on the effects of economic uncertainty on Chinese firms in light of the top-down control by the government, which contrasts with other major economies. This article aims to examine two research questions: 1. How are the capital structures of Chinese listed firms affected by high degrees of economic uncertainty? 2. Are there differences in the impact of such uncertainty between state-owned and private-sector firms?

This article may contribute to the literature from three perspectives. First, it analyzes how economic uncertainty affected the capital structure of publicly traded companies in China during 2006–2020, with particular attention to the Great Recession, the Trade War, and the pandemic. Second, considering the distinct nature of the financial system in China, with the potentially heterogeneous effect of economic uncertainty, the sample firms were classified into two categories: private and state-owned enterprises (SOEs). In China, SOEs and non-SOEs have different degrees of access to finance and transaction costs in the capital markets, and distinct forms of information asymmetry (Wang and Deng 2006). Third, while China has made the transition from a planned to a market economy during the study period, its economic structure arguably remains different from that of other major economies, and thus evidence from China could be of value.

This paper collected fundamental data on A-share firms in China traded on the Shanghai Stock Exchange (SHSE) and Shenzhen Stock Exchange (SZSE). Panel econometric models were employed (Tsusaka and Otsuka 2013), including several covariates that potentially influenced capital structures, such as asset tangibility, profitability, and firm size (Demirgüç-Kunt et al. 2020). The rest of the paper is organized as follows. Relevant literature is reviewed in the next section to delineate the research gap and formulate hypotheses to be tested. The research method is explained in Section 3, while the findings are described and discussed in Section 4. Section 5 concludes the paper.

## 2. Economic Uncertainty and Capital Structure: A Review

The literature elaborates on how macroeconomic uncertainty causes dynamic changes in the capital structures of publicly traded firms, especially in developed countries. Iqbal and Kume (2014) examined the impacts of the Great Recession in 2008 on the debt ratios of firms in the United Kingdom, France, and Germany. Recent research on firms' liquidity in the USA and other developed markets showed negative connections between economic uncertainty and debt levels, liquidity, and investments (D'Mello and Toscano 2020). A

notable exception is Fidrmuc and Korhonen (2010), who investigated emerging markets and identified the impacts of the Great Recession on the economic cycles in China and India. Furthermore, several studies assessed how EPU affected corporate capital structure, governance, investment choices, and risk-taking (Gulen and Ion 2015; D'Mello and Toscano 2020). Istiak and Serletis (2020) explored the connection between uncertainty and firm leverage via six uncertainty indices for financial firms based in the USA. They found that greater risk and uncertainty decreased the real GDP through the effects on consumption, investment, export, and import and that broker–dealers' and shadow banks' total assets fell when the real GDP and asset values declined.

Disruptions in economic environments cause changes in demand and supply, accompanying market fluctuations. In addition, external shocks such as pandemics and geopolitical conflicts can have a massive effect on financial markets and the real economy (Chiang 2021). Previous research suggests connections between EPU and firms' risk-taking (Chatjuthamard et al. 2019) and investments (Gilchrist et al. 2014; Gulen and Ion 2015).

### 2.1. Determinants of Capital Structure

In general, corporate capital structure can be characterized by the proportion of debt and equity, which represents the debt-paying ability of firms (Berk and DeMarzo 2017). Capital structure evolved as a major topic in the area of corporate finance since the pioneering work by Modigliani and Miller (1958), proposing that, in a frictionless environment, corporate capital structure has no impact on the firm's value, which is referred to as the MM theorem. In reality, however, the existence of tax and transaction costs makes markets frictional. Accordingly, the MM theorem was modified to incorporate the effects of corporate taxes (Miller 1977). As a result, a linear connection was found between the firm value and the leverage ratio due to the tax exemption on interest payments. The MM theorem was supported by other prominent scholars such as Smith (1972) and Kraus and Litzenberger (1973), who posited that under a relatively relaxed assumption, there is an ideal capital structure that incorporates the trade-off between debt tax benefits and the expenses of bankruptcy.

Multiple other theories have been developed over the last several decades in an attempt to better understand how businesses decide capital structures. For a long time, several core capital structure theories dominated the literature, including the trade-off theory (Kraus and Litzenberger 1973), the pecking order theory (Myers and Majluf 1984), and the market-timing theory (Rajan and Zingales 1995), among others. Firms' information environment also garnered a great deal of attention, especially the influence of quality of information (Bharath et al. 2009).

Several empirical studies have been conducted to determine which factors had significant impacts on firms' capital structure. Asset tangibility, company size, profitability, growth, liquidity, and other firm-level factors were among the most common variables identified by this line of research (Hundal et al. 2018; Titman and Wessels 1988; Rajan and Zingales 1995; Myers and Majluf 1984; Im et al. 2020). The maturity of corporate debt also exhibits a key role since it is related to rollover risks, thereby affecting operations and risk control, and avoiding corporate bankruptcy (Demirgüç-Kunt et al. 2020).

The majority of research on capital structures comes from developed markets. For example, Rajan and Zingales (1995) investigated the capital structures of listed companies in the G7 nations and found that factors influencing capital structures in the US were the same as in the other G7 states. In contrast, emerging markets typically differ from developed markets in terms of regulations, financial systems, and firm ownership structure, especially in China where shareholdings are concentrated (Li 2020). As a result of the financial liberalization, the Chinese market gradually became more associated with the global markets (Huang and Song 2006). Yet, the capital structure may show different adjustments in response to economic uncertainty as the choices and limitations faced by Chinese firms vary significantly from those in the West (Jiang et al. 2020). Chen (2004) used panel data of 77 Chinese-listed companies from 1994 to 2000 to examine the drivers of corporate capital

structure; they found that certain current finance theory ideas are adaptable to China in that China has some of the same firm-specific characteristics that Western nations use to explain capital structure. Qian et al. (2009) explored whether Chinese enterprises adapted their capital structures to meet a target level and how economic developments in China influenced listed firms' capital structures and found that due to the less developed local bond markets, Chinese firms held relatively high levels of cash reserves, leading to higher agency costs but lower bankruptcy risk.

### 2.2. Economic Uncertainty and Capital Structure

It is not a new notion that economic uncertainty influences corporate capital structure. The definition of uncertainty provided by Milliken (1987) is "perceived incompetence to forecast something precisely", and it is recognized as an environmental variable that affects the whole industry or economy, instead of a single firm. Distinct from risk, uncertainty is multidimensional and thus difficult to predict, as it can originate from various sources such as unpredictable monetary expansion by central banks, real estate price bubbles, and geopolitical events (e.g., Brexit) (Cortes et al. 2021; Jarrow and Silva 2015; Born et al. 2019), and the effects of uncertainty can be global (Alpers 2019). Although most managers are capable of estimating business risks, outcomes of uncertainty are more difficult to deal with because of the broader impact as well as the chain effects (Sharma et al. 2020).

Additionally, the US Federal Reserve Board (FRB) and other major central banks performed several rounds of monetary expansion from the start of the Great Recession to the period of the COVID-19 pandemic (Cortes et al. 2021). According to various studies (Dedola et al. 2020; Cortes et al. 2021; Hattori et al. 2016), quantitative easing rounds influence the capital structure globally by affecting currency rates and asset prices.

Economic uncertainty affects firms in several ways, such as investment decisions, financing, and liquidity (He et al. 2020). According to the pecking order theory, when companies fund new initiatives, they prioritize the use of internal cash surplus, followed by debt financing, and lastly, equity financing. Using bank-level panel data from around 500 commercial banks in seven rising Asian nations, Wu et al. (2021) found that greater economic uncertainty slows loan growth, narrows interest rate spreads, and raises risk, while also encouraging banks to boost capital holdings. Therefore, when facing inadequate funds during the economic uncertainty, firms tend to postpone investments as the future cash flows of new projects become more difficult to predict, which in turn leads to lower demand for debt financing.

Increased economic uncertainty can also impact corporate capital structures due to the accompanying increase in firms' business uncertainty and risk, as well as the decline in the expected stock return. The 2008 Great Recession is an example that led to a sharp increase in subprime mortgage defaults as well as the shortage of credit supply. Consistent with this argument, other scholars report a rapid decline in lending from banks with constrained liquidity (Cornett et al. 2011; Ivashina and Scharfstein 2009). Before, during, and after the Great Recession, researchers examined how stock market uncertainty in the USA influenced stock returns in Latin America and other developing markets and concluded that the heightened uncertainty produced dents in emerging market returns (Sarwar and Khan 2016). Likewise, Byrne et al. (2015) found a negative impact of uncertainty on the performance of US nonfinancial firms, specifically lower expected stock returns, return on equity, and profitability (Jia and Li 2020). Byrne et al. (2015) also found that during times of high economic uncertainty, British firms face higher bankruptcy probability due to increased systemic risk, especially during a financial crisis.

Economic uncertainty can also affect the maturity structure of corporate debt. If expected returns decline, lenders and borrowers are less willing to rely on long-term debt. Thus, short-term debt becomes more appealing (Gürkaynak and Wright 2012; Dick et al. 2013), and the percentage of short-term debt in corporate capital structures might rise. However, evidence from Vermoesen et al. (2012) also indicates that firms with a relatively

higher percentage of short-term debt are less likely to maintain their level of investment during times of high uncertainty, leading to diminishing total debt levels.

In China, Zhang et al. (2015) investigate the impacts of EPU on the corporate capital structure of publicly traded firms through supply and demand effects in 2003–2013. They find that Chinese firms tend to decrease their leverage ratios when EPU rises. However, the effect of uncertainty on leverage is less for SOEs and other types of firms with more robust ties with banks. Directly related, an empirical study by Liu and Zhang (2020) finds that EPU substantially impedes real investment and decreases net debt issuance for private Chinese companies but has no impact on SOEs. They adopt a difference-in-difference approach centered around the supply-side structural reform in 2015 and report a negative effect of EPU on firms' debt issuance, especially on short-term debt. Finally, Li (2020) estimates firm-level uncertainty for Chinese firms using regression analysis of stock returns and an EPU index, finding that higher policy risk was associated with lower leverage in the period 2002–2018. Li and Qiu (2022) find that equities market in China is what transfers EPU shocks to the capital structure and illustrate the negative relationship between EPU and leverage. Chen et al. (2014) analyze stock return volatility as the proxy of a source of uncertainty and demonstrate that firms with significant return volatility lower their leverage ratios the next year.

### 2.3. Contribution to Literature

The present study makes three distinct contributions to the literature. Following the approach by Zhang et al. (2015), the data extend to cover the periods of the US–China Trade War and the COVID-19 pandemic, during which uncertainty was arguably substantially elevated. This allows us to examine whether the impact of those recent high-uncertainty events on firm leverage was different from the Great Recession case, which occurred when China's economy was still in transition and not yet as integrated with the global economy as it is today. In addition, panel regression methods were employed to control for time-invariant, firm-specific omitted variables. Moreover, existing studies have mainly investigated the impact of EPU (Zhang et al. 2015; Li et al. 2019; Chen and Chiang 2020), while ours applied the more generalized WUI (world uncertainty index) for China as a broader measure of economic uncertainty, as this article focuses on global uncertainty rather than domestic policy changes.

### 3. Hypothesis Formulation

Firms' capital structure and decision-making are inextricably linked to the external economic environment. Economic uncertainty poses threats to firms' operations and profitability, as it can lead to lower demand for firm products, delayed investments, an increase in cash retention, and a decline in issues of debt and equity (Colak et al. 2017). Brunnermeir and Oehmke (2013) demonstrated that if companies prioritize financial flexibility in times of economic uncertainty, they are less inclined to engage in long-term businesses. During a crisis, both banks and investors become hesitant to lock in money in long-term projects as uncertainty and risk increase while anticipated returns fall (Dick et al. 2013). Therefore, firms that are not capable of meeting the requirement of debt repayment may opt to decrease their loan maturity and leverage when uncertainty increases and business prospects grow more unclear.

The stability of the operating environment affects corporate strategies and investment decisions (Rodriguez Lopez et al. 2017) and hereby further affects corporate borrowing, leading to the adjustment of the debt and equity proportions. The increasing volatility of expected returns would make capital providers exposed to higher risk, especially in countries with high bankruptcy costs and monitoring costs (Chen and Funke 2009). In an environment with higher uncertainty, firms are more likely to put off their decisions on investment or M&A (mergers and acquisitions), implementing those projects when the economic environment is more stable and predictable (Stokey 2016). The reduction in

investments leads to lower demand for external finance and thus borrowing, thereby also leading to lower firm leverage. Therefore, the following hypothesis is proposed:

**H$_1$.** *Economic uncertainty has negative impacts on the leverage of Chinese listed firms.*

China's institutional systems are distinct from those of both industrialized and developing countries. The institutional framework for Chinese companies has two distinguishing characteristics. First, China is transitioning from a planned to a market economy. Second, many Chinese listed firms were originally founded by the government, and the government often retains a controlling position in these companies after they go public. According to the China Banking and Regulatory Commission (CBRC), to qualify for long-term loans, firms must be involved in industrial projects, land development, environmental protection, or long-term investment projects. SOEs have more opportunities to engage in such projects, and thus their long-term debt ratios are predictably greater than those of non-SOEs (Liu and Zhang 2020). Studies have also argued that Chinese banks prefer to lend to SOEs (Brandt and Li 2003; Firth et al. 2008; Cull et al. 2015), most likely due to their better connections and lower bankruptcy costs. Hence, the following hypothesis is formulated:

**H$_2$.** *The debt level of SOEs is less affected by economic uncertainty than that of non-SOEs.*

Apart from the general impact of economic uncertainty as proxied by the WUI for China, this article also addresses the impact of the following three global events with high uncertainty: the Great Recession from 2008 to 2009, the US–China Trade War in 2019, and the COVID-19 pandemic in 2020. China's economic shift in the last two decades is marked by the shift in financial systems from one dominated by banks and relationships to one dominated by securities markets at arm's length (Alon et al. 2014). Such transitions attract increased foreign direct investments and build stronger connections with the global markets, implying that the Chinese market is increasingly exposed to global economic uncertainty. On this ground, two more hypotheses are presented as follows:

**H$_3$.** *The capital structure of Chinese listed firms was affected differently by the 2008 Great Recession, the US–China Trade War, and the COVID-19 pandemic.*

**H$_4$.** *The influences of the three crises on debt ratios are weaker for SOEs than for non-SOEs.*

## 4. Data and Methods

The empirical analysis draws on annual data from the Chinese Stock Markets and Accounting Research (CSMAR) database. The panel data cover the financial statements from 2006 to 2020 for all firms listed in the Shanghai and Shenzhen Stock Exchange markets. This research chooses 2006 as the starting year to cover the period just before the Great Recession, also because key data are missing prior to 2006. B-share firms were excluded because the issuance of B-shares follows different regulations and financing access. This resulted in the data coverage of 1138 unique firms. In the statistical analysis and verification of hypotheses, panel regression models were employed to minimize the estimation bias arising from endogeneity and omitted firm-level variables. Furthermore, a robustness test was conducted by using Merton's distance to default as an instrumental variable to correct for possible endogeneity in firm return.

*4.1. Sample and Variables Construction*

The purpose of the regression models is to test the relation between economic uncertainty and corporate capital structures while controlling for other relevant variables that affect capital structures in order to isolate the effect of uncertainty. To represent firm leverage, this article uses three dependent variables: the overall, long-term, and short-term debt-to-assets ratios. Table A1 in Appendix A lists and defines the key variables included in the analytical models. For debt-to-asset ratios, this article applied the book leverage ratio, which is defined as the book value of debt divided by total assets. As the key independent variable, the WUI for China was included to quantify the influence of economic uncertainty on capital structures. The WUI was developed by Ahir et al. (2018), which is calculated by counting the number of times the term "uncertain" (or a variation of it) appears in country reports from the Economist Intelligence Unit and is then rescaled by a factor of one million. The WUI measures economic uncertainty at the global level, as well as at the regional and country levels. The WUI for China is the version specific to China.

Due to the tight substitutability of corporate and government bonds, higher governmental deficits can have an impact on capital structure (Demirci et al. 2019; Silva 2020). Therefore, the analysis also considers the fiscal situation of the Chinese government, using the data on percentage of net lending out of the GDP.[1] Brexit was a major event that created global economic uncertainty, particularly for firms that had business with the UK (Campello et al. 2022). During the same period, trade and diplomatic ties between China and Brazil were weakened, hurting major exporters of goods. Hence, the EPU indices of the UK[2] and Brazil[3] are included as control variables.

As an alternative proxy for uncertainty, dummy variables were introduced for three events that had a global impact and caused high economic uncertainty. First, the 2008–2009 Great Recession was represented by the dummy that took the value of one in the years 2008 and 2009 and zero otherwise. Nevertheless, to capture the impact of quantitative easing during the Great Recession, the monetary expansion by the FRB is added to the regression model, measured as Central Bank Assets to GDP[4]. Second, the US–China Trade War period was represented by a dummy variable equal to one in the years 2018 and 2019. The third event was the COVID-19 pandemic, with a corresponding dummy variable equal to one in the year 2020. The three event variables aim to verify whether the impacts on leverage differed with the consecutive development of the Chinese economy.

Empirical studies suggest that firm size is positively related to leverage (Huang and Song 2006; Friend and Lang 1988). This research applies firm size as one of the covariates. Commonly, firm size is quantitatively valued via the logarithm of total assets to address the right skew in the distribution. According to the trade-off theory, profitable companies are more likely to borrow money and have a higher level of debt in their capital structure due to the interest tax shield (Kraus and Litzenberger 1973). To represent firms' profitability, firm stock return and return on assets (ROA) were included. Income growth was used as a proxy for future growth potential as the future expectation affects the current capital structure decision (Firth et al. 2008).[5] The values of ROA and income growth were Winsorized at 5% and 95% to reduce the influence of extreme values.

Another variable related to leverage was the proportion of tangible assets out of total assets. Tangible assets may facilitate external borrowing from banks or other financial institutions, as tangible assets are easier to value and more difficult to expropriate. Furthermore, a dummy variable was included to indicate special treatment (ST) firms. The China Securities Regulatory Commission (CSRC) designates ST to publicly-traded firms that have had "abnormal financial situations" or "other scenario abnormalities".[6] The daily stock price fluctuation of ST firms is restricted to a maximum of 5%. Interbank borrowing dominates the Chinese bond market, rather than bond issuance by industrial or service companies, as corporate bond markets are relatively underdeveloped (Michael et al. 2019). SOEs have easy access to debt financing through state-owned banks, which supply fund to the majority of SOEs. Therefore, to distinguish between SOEs and non-SOEs, a dummy variable for state ownership was included in the analysis.

### 4.2. Empirical Models

The hypotheses concerning how economic uncertainty impacts the capital structure of Chinese-listed companies were tested via panel regression models. The Hausman test was applied to determine between the fixed effect and random effect models (Hausman 1978; Tsusaka and Otsuka 2013). The results of the test were shown in each of the regression tables. Moreover, the standard errors were clustered by firms and years to account for the correlation in error terms within firm-specific time series and year-specific clusters.

Hypothesis 1 was tested through the following model specification:

$$\text{Leverage}_{it} = \alpha_i + \beta_1 \text{WUI\_China}_t + \beta_2 \text{TA\_GDP\_US}_t + \beta_3 \text{Fiscal\_China}_t + \beta_4 \text{EPU\_UK}_t + \\ \beta_5 \text{EPU\_Brazil}_t + \beta_6 \text{SOEs\_Dum}_{it} + \beta_7 \text{ST\_Dum}_{it} + \sum_j \beta_j X_{ijt} + \varepsilon_{it} \tag{1}$$

where i represents each firm and t represents each year. Leverage represents either the overall book leverage, long-term book leverage, or short-term book leverage. The market leverage ratio was not included because, before 2007, a significant portion of Chinese listed firms' shares were non-tradable. The independent variables were as shown in Table A1. $\alpha_i$ is unobservable time-invariant firm-specific effects. This term characterizes panel regression models and absorbs the firm-level heterogeneity that would otherwise cause estimation biases, particularly the endogeneity bias. The Hausman test examines whether $\alpha_i$ is correlated with the independent variables. When correlated, the random effect model is inconsistent and thus the fixed effect model is appropriate, as the latter is always consistent. When uncorrelated, the random effect model is also consistent and can be used, as it is more efficient.

The government owns the majority of commercial banks and state-owned businesses, which is the distinct characteristic of the Chinese economic system. Because of their political ties, state-owned businesses are often afforded first priority in receiving commercial bank loans. At the same time, bank loan issuance is less sensitive to default risk for SOEs, due to implicit government backup. To test Hypothesis 2, an interaction term between the WUI China and firm ownership was added into the regression model as follows:

$$\text{Leverage}_{it} = \alpha_i + \beta_1 \text{WUI\_China}_t + \beta_2 \text{SOEs\_Dum}_{it} + \beta_3 \text{SOEs\_Dum}_{it} \times \text{WUI\_China}_t + \\ \beta_4 \text{TA\_GDP\_US}_t + \beta_5 \text{SOEs\_Dum}_{it} \times \text{TA\_GDP\_US}_t + \beta_6 \text{Fiscal\_China}_t + \beta_7 \text{SOEs\_Dum}_{it} \times \\ \text{Fiscal\_China}_t + \beta_8 \text{EPU\_UK}_t + \beta_9 \text{SOEs\_Dum}_{it} \times \text{EPU\_UK}_t + \\ \beta_{10} \text{EPU\_Brazil}_t + \beta_{11} \text{SOEs\_Dum}_{it} \times \text{EPU\_Brazil}_t + \beta_{12} \text{ST\_Dum}_{it} + \sum_j \beta_j X_{ijt} + \varepsilon_{it} \tag{2}$$

As an alternative to the WUI-based specification, models with three dummies representing the major economic crises were utilized. The impacts of the three events—which symbolize high economic uncertainty periods—on firm leverage was examined using the following specification:

$$\text{Leverage}_{it} = \alpha_i + \beta_1 \text{Crisis\_Dum}_t + \beta_2 \text{TW\_Dum}_t + \beta_3 \text{Covid\_Dum}_t + \beta_4 \text{SOE\_Dum}_{it} + \\ \beta_5 \text{TA\_GDP\_US}_t + \beta_6 \text{Fiscal\_China}_t + \beta_7 \text{EPU\_UK}_t + \beta_8 \text{EPU\_Brazil}_t + \beta_9 \text{ST\_Dum}_{it} + \\ \sum_j \beta_j X_{ijt} + \varepsilon_{it} \tag{3}$$

where $\text{Crisis\_Dum}_t$ $\text{TW\_Dum}_t$ and $\text{COVID\_Dum}_t$ are the dummy variables for the respective periods of high economic uncertainty.

In addition, to test the expected lower impact of uncertainty on state-owned firms, interaction terms of state ownership with the three events dummies were included as follows:

$$\text{Leverage}_{it} = \alpha_i + \beta_1 \text{SOEs\_Crisis\_Dum}_{it} + \beta_2 \text{SOEs\_TW\_Dum}_{it} + \\ \beta_3 \text{SOEs\_COVID\_Dum}_{it} + \beta_4 \text{SOEs\_Dum}_{it} + \beta_5 \text{TA\_GDP\_US}_t + \beta_6 \text{SOEs\_Dum}_{it} \times \\ \text{TA\_GDP\_US}_t + \beta_7 \text{Fiscal\_China}_t + \beta_8 \text{SOEs\_Dum}_{it} \times \text{Fiscal\_China}_t + \beta_9 \text{EPU\_UK}_t + \\ \beta_{10} \text{SOEs\_Dum}_{it} \times \text{EPU\_UK}_t + \beta_{11} \text{EPU\_Brazil}_t + \beta_{12} \text{SOEs\_Dum}_{it} \times \text{EPU\_Brazil}_t + \\ \beta_{13} \text{ST\_Dum}_{it} + \sum_j \beta_j X_{ijt} + \varepsilon_{it} \tag{4}$$

Furthermore, while the inclusion of fixed effects lessens endogeneity bias arising from time-invariant, firm-specific unobservable variables, other sources of endogeneity remain. The robustness check uses the panel instrumental variable (IV) method (Wooldridge 2005), where distance to default, as per the Merton model, was added as an IV for correct for possible endogeneity in firm return. The diagnostic F statistic was greater than 10 in all models, indicating the relevance of the instrument. The calculation of distance to default is shown in Table A1.

## 5. Results and Discussion

### 5.1. Descriptive Statistics

The descriptive statistics for the main variables are shown in Table 1. The average debt-to-assets ratio was 0.220, while the median was 0.168, which implies that leverage was skewed to the right. It is worth mentioning that short-term leverage was substantially larger than long-term leverage. One possible explanation for this phenomenon is that in China, listed firms with the ability to take on long-term obligations are limited (Jiang et al. 2020). The SOE dummy indicates that 66% of the sample firms were SOEs.

**Table 1.** Descriptive statistics for the variables analyzed.

| Variable | N | Mean | Median | SD | Min | Max |
|---|---|---|---|---|---|---|
| TD/TA | 17,067 | 0.220 | 0.168 | 3.539 | 0.000 | 25.698 |
| LD/TA | 17,067 | 0.068 | 0.023 | 0.100 | 0.000 | 0.846 |
| SD/TA | 17,067 | 0.152 | 0.087 | 3.539 | 0.000 | 25.698 |
| WUI_China | 17,070 | 0.122 | 0.097 | 0.106 | 0.000 | 0.354 |
| TA_GDP_US | 17,070 | 10.535 | 10.890 | 4.676 | 3.235 | 22.420 |
| Fiscal_China | 17,070 | −2.306 | −1.144 | 2.654 | −9.718 | 0.059 |
| EPU_UK | 17,070 | 146.869 | 139.120 | 56.260 | 54.565 | 289.139 |
| EPU_Brazil | 17,070 | 175.362 | 148.635 | 75.467 | 92.777 | 346.490 |
| Firm_Return | 16,947 | 0.324 | 0.078 | 0.897 | −0.866 | 21.53 |
| Tangibility | 17,045 | 0.676 | 0.694 | 0.282 | 0.000 | 1.000 |
| Firm_Size | 17,068 | 22.12 | 22.19 | 2.096 | 10.57 | 28.26 |
| ROA | 17,039 | 0.043 | 0.038 | 0.047 | −0.055 | 0.147 |
| IG | 16,936 | 0.344 | 0.118 | 0.721 | −0.399 | 2.675 |
| ST_Dum | 17,070 | 0.053 | 0.000 | 0.224 | 0.000 | 1.000 |
| SOEs_Dum | 17,070 | 0.656 | 1.000 | 0.475 | 0.000 | 1.000 |
| SOEs_WUI | 17,070 | 0.080 | 0.013 | 0.104 | 0.000 | 0.354 |
| Crisis_Dum | 17,070 | 0.133 | 0.000 | 0.340 | 0.000 | 1.000 |
| Tradewar_Dum | 17,070 | 0.133 | 0.000 | 0.340 | 0.000 | 1.000 |
| COVID_Dum | 17,070 | 0.067 | 0.000 | 0.249 | 0.000 | 1.000 |
| SOEs_Crisis | 17,070 | 0.090 | 0.000 | 0.286 | 0.000 | 1.000 |
| SOEs_TW | 17,070 | 0.085 | 0.000 | 0.279 | 0.000 | 1.000 |
| SOEs_COVID | 17,070 | 0.043 | 0.000 | 0.203 | 0.000 | 1.000 |

Note: Some variables have a few missing observations. Thus, the numbers of observations are not exactly the same across variables.

### 5.2. The Impacts of the WUI on Firm Leverage

Table 2 shows the results of the panel regression estimates. The WUI index of China had negative effects on firms' total leverage, which supports Hypothesis 1. As the WUI for China increased by one point, total leverage decreased by 0.084 on average, holding all the covariates unchanged. Other uncertainty proxies showed the same trend with the WUI, though the impacts were minimal. This finding is consistent with Graham et al. (2015) and Li and Qiu (2022) that during periods of economic recession and times of uncertainty, opportunities for investments decrease and demand for external capital shrinks, resulting in a reduction in firms' leverage ratios. Furthermore, the negative impact of uncertainty was larger for short-term debt than for long-term debt, which may be interpreted by the ease with which firms can modify the quantity of short-maturity debt.

**Table 2.** The impacts of economic uncertainty on corporate capital structure.

| Variable | TD/TA | LD/TA | SD/TA |
|---|---|---|---|
| WUI_China | −0.0842 *** | −0.0104 | −0.0468 * |
| | (0.0113) | (0.00840) | (0.0243) |
| TA_GDP_US | −0.00346 *** | −0.000776 *** | −0.00284 *** |
| | (0.000247) | (0.000174) | (0.000464) |
| Fiscal_China | 0.00297 *** | 0.00320 *** | −0.000609 |
| | (0.000495) | (0.000360) | (0.000834) |
| EPU_UK | −0.0000846 *** | 0.0000278 * | −0.000111 ** |
| | (0.0000202) | (0.0000145) | (0.0000449) |
| EPU_Brazil | −0.000129 *** | −0.0000701 *** | −0.00000557 |
| | (0.0000169) | (0.0000118) | (0.000047) |
| SOEs_Dum | 0.00919 * | 0.00650 * | −0.0835 * |
| | (0.00519) | (0.00333) | (0.0432) |
| ST_Dum | 0.0334 *** | −0.00776 ** | 0.130 *** |
| | (0.00527) | (0.00362) | (0.0237) |
| Firm_Return | 0.00716 *** | 0.00481 *** | −0.00232 |
| | (0.00120) | (0.000812) | (0.00346) |
| Tangibility | 0.0639 *** | 0.0322 *** | 0.0613 *** |
| | (0.00697) | (0.00492) | (0.0223) |
| Firm_Size | 0.0263 *** | 0.0235 *** | −0.0282 ** |
| | (0.00144) | (0.000953) | (0.0141) |
| ROA | −0.611 *** | −0.177 *** | −0.679 *** |
| | (0.0225) | (0.0145) | (0.133) |
| IG | −0.000727 | 0.00265 *** | −0.00361 |
| | (0.00135) | (0.000993) | (0.00247) |
| Model | FE | FE | FE |
| Hausman test (*p*-value) | 0.0000 | 0.0000 | 0.0000 |
| Num. of observations | 16,847 | 16,847 | 16,847 |
| Num. of firms | 1138 | 1138 | 1138 |
| $R^2$ | 0.148 | 0.069 | 0.047 |

Note: ***, **, and * indicate statistical significance at 1%, 5%, and 10% levels, respectively. Standard errors are in the parentheses.

Although the ratio of total assets to the GDP in the US and the EPU in the UK and Brazil were negatively associated with firm leverage, the magnitude of the coefficient was small, indicating that the impact was minimal. In contrast, as the rate of net lending by the Chinese government increased by one percentage point, the total leverage and long-term leverage rose by 0.0029.

Regarding the control variables, firm size was positively associated with debt to asset ratio and long-term leverage, indicating that large firms preferred long-term loans. From another perspective, banks assign priority to large firms since large firms generally have more stable income and lower solvency risk. Tangibility was substantially positively related with firm leverage. This result is consistent with Bhabra et al. (2008) and Zou and Xiao (2006), who reported positive relationships between tangibility and leverage, which is also in line with the prediction made using the trade-off theory that businesses with greater tangible assets have fewer agency problems and lower distress costs. Furthermore, the result revealed a negative link between ROA and firm leverage, which supports the pecking order hypothesis and capital structure research regarding China. Income growth was positively associated with long-term leverage. However, it did not show a significant impact on total and short-term leverage, which is also consistent with the findings of Huang and Song (2006). Special treatment had positive effects on total leverage and long-term leverage, while state ownership had negative and positive effects on long-term and short-term leverage, respectively.

Table 3 presents the estimations including the interaction terms of the WUI and state ownership. The coefficients on the interaction term indicate that the effect of WUI differed

by state ownership on total leverage and short-term leverage by not on long-term leverage. Specifically, on total leverage, the marginal effect of the WUI was −0.0692 (i.e., −0.118 + 0.0488) for SOEs and −0.118 for non-SOEs. In other words, as the WUI increased by one point, total leverage decreased by 0.0692 for SOEs and by 0.118 for non-SOEs, suggesting that non-SOEs were more sensitive to economic uncertainty than SOEs. Likewise, on short-term leverage, the marginal effect of the WUI was −0.0247 for SOEs and −0.0983 for non-SOEs. In contrast, the marginal effect on long-term leverage was not significant, regardless of state ownership. Hence, Hypothesis 2 was upheld for total and short-term leverage but not confirmed for long-term leverage.

**Table 3.** The impacts of the economic uncertainty on capital structure by state ownership.

| Variable | TD/TA | LD/TA | SD/TA |
|---|---|---|---|
| WUI_China | −0.118 *** | −0.0121 | −0.0983 *** |
| | (0.0205) | (0.0144) | (0.0347) |
| SOEs_Dum | −0.0205 *** | 0.00913 * | −0.136 *** |
| | (0.00760) | (0.00507) | (0.0431) |
| SOEs_Dum × WUI_China | 0.0488 ** | 0.00267 | 0.0736 ** |
| | (0.0244) | (0.0177) | (0.0341) |
| TA_GDP_US | −0.00408 *** | −0.000771 *** | −0.00541 *** |
| | (0.000421) | (0.000283) | (0.00108) |
| SOEs_Dum × TA_GDP_US | 0.000885 * | 0.000000591 | 0.00380 *** |
| | (0.000500) | (0.000349) | (0.00135) |
| Fiscal_China | 0.00190 ** | 0.00307 *** | −0.00281 ** |
| | (0.000858) | (0.000569) | (0.00140) |
| SOEs_Dum × Fiscal_China | 0.00159 | 0.000191 | 0.00324 ** |
| | (0.00100) | (0.000690) | (0.00165) |
| EPU_UK | −0.000127 *** | 0.000029 | −0.000208 *** |
| | (0.0000336) | (0.0000244) | (0.0000622) |
| SOEs_Dum × EPU_UK | 0.0000645 | −0.00000191 | 0.000147 ** |
| | (0.0000399) | (0.0000288) | (0.0000734) |
| EPU_Brazil | −0.000164 *** | −0.0000618 *** | 0.0000274 |
| | (0.0000289) | (0.0000196) | (0.0000889) |
| SOEs_Dum × EPU_Brazil | 0.0000525 | −0.0000126 | −0.0000533 |
| | (0.0000352) | (0.0000243) | (0.0000901) |
| ST_Dum | 0.0324 *** | −0.00765 ** | 0.129 *** |
| | (0.00526) | (0.00363) | (0.0236) |
| Firm_Return | 0.00727 *** | 0.00479 *** | −0.00222 |
| | (0.00119) | (0.000813) | (0.00350) |
| Tangibility | 0.0655 *** | 0.0320 *** | 0.0642 *** |
| | (0.00696) | (0.00490) | (0.0226) |
| Firm_Size | 0.0268 *** | 0.0234 *** | −0.0274 * |
| | (0.00144) | (0.000960) | (0.0141) |
| ROA | −0.608 *** | −0.177 *** | −0.675 *** |
| | (0.0224) | (0.0145) | (0.134) |
| IG | −0.000891 | 0.00268 *** | −0.00395 |
| | (0.00135) | (0.000994) | (0.00245) |
| Model | FE | FE | FE |
| Hausman test (*p*-value) | 0.0000 | 0.0000 | 0.0000 |
| Num. of observations | 16,847 | 16,847 | 16,847 |
| Num. of firms | 1138 | 1138 | 1138 |
| $R^2$ | 0.150 | 0.069 | 0.049 |

Note: ***, **, and * indicate statistical significance at 1%, 5%, and 10% levels, respectively. Standard errors are in the parentheses.

The impact of the US FRB's quantitative easing differed significantly between SOEs and non-SOEs, though the magnitude was small. The marginal effect of the TA_GDP_US was approximately −0.0032 for SOEs and −0.0041 for non-SOEs. In contrast, the influence of the fiscal situation of China, EPU_UK, and EPU_Brazil did not appear to differ significantly between SOEs and non-SOEs. Since SOEs have political goals, such as supporting employment and economic growth, it is likely that if they undergo financial difficulties, the government may assist them with loans or reduced taxes. Previous loans may also be written off or the conditions may be adjusted by the government (Zhengwei 2013). In another aspect, banks are more inclined to lend money to larger firms than smaller firms, especially when risks arise (Qian et al. 2009). SOEs in China tend to be very diversified, and therefore less likely to fail on their loan obligations. Similarly, Liu and Zhang (2020) found that non-SOEs in China reduced their short-term debt issuance to a greater extent than SOEs during times of high uncertainty.

*5.3. Impacts of the Specific Economic Crises on Firm Leverage*

Table 4 shows that the primary explanatory variables of interest, namely the three dummies representing the major economic crises, i.e., the 2008 Great Recession, the US–China Trade War, and the COVID-19 pandemic, generally had statistically significant coefficients. The total debt ratios of Chinese firms decreased on average during the Great Recession, while the long-term leverage slightly increased with the marginal effect of 0.005. The total leverage and long-term leverage also declined during the Trade War period. In contrast, the COVID-19 dummy was positively associated with total leverage and short-term leverage. The obtained evidence is basically in support of Hypothesis 3.

Table 5 shows that only three out of the seven interaction terms were statistically significant. Specifically, the effects of the Trade War, the monetary expansion, and Brexit on total leverage differed by state ownership of firms, where the Trade War had positive effects (0 + 0.019) among SOEs and insignificant effects among non-SOEs, whilst the pandemic had larger positive effects (0.0499) among non-SOEs than among SOEs (−0.0118 = −0.0118 + 0.00499). On total leverage and long-term leverage, the effects of the three economic crises did not differ by state ownership of firms. On any type of leverage, the effects of the Great Recession did not differ by state ownership. The results suggest that SOEs lowered their short-term leverage to a lesser extent during the COVID-19 pandemic than non-SOEs and that SOEs raised short-term leverage during the Trade War while non-SOEs did not. On the whole, Hypothesis 4 was only weakly supported.

Thus far, empirical research on the capital structure impact of state ownership in China has shown mixed results. Bhabra et al. (2008) and Qian et al. (2009) found a linkage between state ownership and company leverage ratios and observed that SOEs in China utilized less debt financing than non-SOEs. On the other hand, a recent study by Li (2020) argued that firm ownership was not significantly related to firm leverage.

Both Model 3 and Model 4 identified significant decreases in debt financing during the 2008 Great Recession. On the other hand, leverage ratios generally increased in the more recent periods of high uncertainty, in line with Hypothesis 3. Furthermore, Model 4 showed that uncertainty had a lower impact on the short-term leverage of SOEs (compared to non-SOEs) during the COVID-19. This can be explained by SOE's easier access to short-term finance from connected state banks. Moreover, the US–China Trade War may dissuade investment and diminish productivity in both countries due to greater uncertainty and reduced openness (Itakura 2019).

**Table 4.** The impacts of the Great Recession, Trade War, and COVID-19 on capital structure.

| Variable | TD/TA | LD/TA | SD/TA |
|---|---|---|---|
| Crisis_Dum | −0.00897 *** | 0.00489 ** | −0.0102 |
| | (0.00286) | (0.00203) | (0.0101) |
| Tradewar_Dum | −0.0460 *** | −0.0283 *** | 0.00981 |
| | (0.00582) | (0.00400) | (0.0269) |
| COVID_Dum | 0.0424 *** | −0.00268 | 0.0607 ** |
| | (0.0101) | (0.00711) | (0.0286) |
| SOEs_Dum | 0.00319 | 0.00384 | −0.0852 ** |
| | (0.00511) | (0.00333) | (0.0422) |
| TA_GDP_US | −0.00756 *** | −0.00165 *** | −0.00487 *** |
| | (0.000347) | (0.000239) | (0.00114) |
| Fiscal_China | 0.00153 | 0.000115 | 0.00477 |
| | (0.00134) | (0.000922) | (0.00486) |
| EPU_UK | −0.000132 *** | 0.000218 | −0.000123 ** |
| | (0.0000197) | (0.000014) | (0.000059) |
| EPU_Brazil | −0.000108 *** | −0.000123 *** | 0.0000983 |
| | (0.0000.99) | (0.0000140) | (0.0000779) |
| Firm_Return | 0.00239 * | 0.00196 ** | −0.00112 |
| | (0.00143) | (0.000969) | (0.00330) |
| Tangibility | 0.0261 *** | 0.0164 *** | 0.0474 * |
| | (0.00729) | (0.00535) | (0.0253) |
| Firm_Size | 0.0344 *** | 0.0273 *** | −0.0260 * |
| | (0.00153) | (0.00102) | (0.0154) |
| ROA | −0.616 *** | −0.173 *** | −0.691 *** |
| | (0.0223) | (0.0146) | (0.136) |
| IG | −0.00149 | 0.00237 ** | −0.00374 |
| | (0.00134) | (0.000987) | (0.00267) |
| ST_Dum | 0.0345 *** | −0.00690 * | 0.130 *** |
| | (0.00521) | (0.00361) | (0.0233) |
| Model | FE | FE | FE |
| Hausman test (*p*-value) | 0.0000 | 0.0000 | 0.0000 |
| Num. of observations | 16,847 | 16,847 | 16,847 |
| Num. of firms | 1138 | 1138 | 1138 |
| $R^2$ | 0.170 | 0.081 | 0.048 |

Note: ***, **, and * indicate statistical significance at 1%, 5%, and 10% levels, respectively. Standard errors are in the parentheses.

**Table 5.** The impacts of the Great Recession, Trade War, and COVID-19: SOEs vs. non-SOEs.

| Variable | TD/TA | LD/TA | SD/TA |
|---|---|---|---|
| SOEs_Crisis | 0.00819 | 0.00138 | 0.00186 |
| | (0.00620) | (0.00422) | (0.0295) |
| SOEs_TW | 0.0190 * | −0.00527 | −0.0152 |
| | (0.0101) | (0.00669) | (0.0471) |
| SOEs_COVID | −0.0118 | −0.0109 | −0.0970 |
| | (0.0187) | (0.0125) | (0.0834) |
| SOEs_Dum | −0.0408 *** | 0.00554 | −0.141 *** |
| | (0.00936) | (0.00626) | (0.0368) |
| Crisis_Dum | −0.0147 *** | 0.00400 | −0.0115 |
| | (0.00524) | (0.00341) | (0.0295) |
| Tradewar_Dum | −0.0594 *** | −0.0247 *** | 0.0190 |
| | (0.00912) | (0.00578) | (0.0561) |
| COVID_Dum | 0.0499 *** | 0.00462 | 0.125 |
| | (0.0163) | (0.0107) | (0.0789) |
| TA_GDP_US | −0.00956 *** | −0.00181 *** | −0.00913 *** |
| | (0.000587) | (0.000380) | (0.00193) |

**Table 5.** *Cont.*

| Variable | TD/TA | LD/TA | SD/TA |
|---|---|---|---|
| SOEs_Dum × TA_GDP_US | 0.00283 *** | 0.000257 | 0.00620 *** |
| | (0.000679) | (0.000460) | (0.00188) |
| Fiscal_China | 0.0000302 | 0.000813 | 0.00862 |
| | (0.00212) | (0.00135) | (0.0118) |
| SOEs_Dum × Fiscal_China | 0.00213 | −0.00102 | −0.00593 |
| | (0.00234) | (0.00156) | (0.0111) |
| EPU_UK | −0.000179 *** | 0.0000227 | −0.000222 ** |
| | (0.0000324) | (0.0000225) | (0.0000906) |
| SOEs_Dum × UK_EPU | 0.0000702 | −0.0000855 | 0.000148 * |
| | (0.000037) | (0.0000264) | (0.0000762) |
| Brazil_EPU | −0.000133 *** | −0.000102 *** | 0.000229 |
| | (0.0000334) | (0.0000219) | (0.000191) |
| SOEs_Dum × EPU_Brazil | 0.0000368 | −0.0000304 | −0.000201 |
| | (0.0000394) | (0.0000267) | (0.000185) |
| Firm_Return | 0.00241 * | 0.00197 ** | −0.00106 |
| | (0.00142) | (0.000971) | (0.00324) |
| Tangibility | 0.0279 *** | 0.0163 *** | 0.0500 ** |
| | (0.00728) | (0.00534) | (0.0253) |
| Firm_Size | 0.0352 *** | 0.0272 *** | −0.0248 |
| | (0.00154) | (0.00104) | (0.0154) |
| ROA | −0.613 *** | −0.173 *** | −0.686 *** |
| | (0.0222) | (0.0146) | (0.137) |
| IG | −0.00181 | 0.00239 ** | −0.00423 |
| | (0.00134) | (0.000988) | (0.00270) |
| ST_Dum | 0.0331 *** | −0.00689 * | 0.128 *** |
| | (0.00518) | (0.00361) | (0.0232) |
| Model | FE | FE | FE |
| Hausman test (*p*-value) | 0.0000 | 0.0000 | 0.0000 |
| Num. of observations | 16,847 | 16,847 | 16,847 |
| Num. of firms | 1138 | 1138 | 1138 |
| $R^2$ | 0.174 | 0.081 | 0.050 |

Note: ***, **, and * indicate statistical significance at 1%, 5%, and 10% levels, respectively. Standard errors are in the parentheses.

## 6. Robustness Check

Table 6 shows the results of the robustness check for Model 1, confirming that the WUI Index of China was negatively related. However, the EPU UK and the net lending rate of the Chinese government were positively related to total and long-term leverage. In contrast, the ratio of assets to the GDP in the US and EPU Brazil were negatively associated with firm leverage.

Table 7 shows the results for Model 2 with the IV. The coefficient of the interaction term between firm ownership and the WUI China was not significant, indicating that the WUI China had the same impact on SOEs and non-SOEs. The coefficient of the interaction term between quantitative easing and firm ownership was approximately 0.003, and the marginal effect on non-SOEs was around −0.001 (i.e., −0.004 + 0.003).

**Table 6.** The impacts of the economic uncertainty on capital structure: robustness check.

| Variable | TD/DA | LD/TA | SD/TA |
|---|---|---|---|
| ROA | −3.756 *** | −1.744 *** | −1.041 |
| | (0.633) | (0.372) | (1.076) |
| WUI_China | −0.0716 *** | −0.00425 | −0.0401 |
| | (0.0204) | (0.0120) | (0.0347) |
| SOEs_Dum | −0.0197 ** | −0.00850 | −0.0868 *** |
| | (0.00922) | (0.00541) | (0.0157) |
| TA_GDP_US | −0.00435 *** | −0.00121 *** | −0.00276 *** |
| | (0.000462) | (0.000271) | (0.000785) |
| Fiscal_China | 0.00728 *** | 0.00537 *** | 0.0000206 |
| | (0.00115) | (0.000677) | (0.00196) |
| EPU_UK | 0.0000822 * | 0.000110 *** | −0.0000836 |
| | (0.0000486) | (0.0000286) | (0.0000827) |
| EPU_Brazil | −0.000195 *** | −0.000102 *** | −0.0000115 |
| | (0.0000335) | (0.0000197) | (0.000057) |
| Firm_Return | 0.0303 *** | 0.0164 *** | 0.000209 |
| | (0.00498) | (0.00293) | (0.00848) |
| Tangibility | −0.0188 | −0.00868 | 0.0486 |
| | (0.0197) | (0.0116) | (0.0335) |
| Firm_Size | 0.0332 *** | 0.0267 *** | −0.0283 *** |
| | (0.00232) | (0.00136) | (0.00395) |
| IG | 0.00638 ** | 0.00602 *** | −0.00177 |
| | (0.00252) | (0.00148) | (0.00429) |
| ST_Dum | 0.0447 *** | −0.0000308 | 0.114 *** |
| | (0.00703) | (0.00412) | (0.0119) |
| Model | FE | FE | FE |
| Hausman test (*p*-value) | 0.0000 | 0.0000 | 0.0000 |
| F-statistic for IV | 56.55 | 45.78 | 45.61 |
| Num. of observations | 16,612 | 16,611 | 16,611 |
| Num. of firms | 1130 | 1130 | 1130 |
| $R^2$ | −1.266 | −0.704 | 0.042 |

Note: ***, **, and * indicate statistical significance at 1%, 5%, and 10% levels, respectively. Standard errors are in the parentheses.

The coefficient of the interaction term between quantitative easing and firm ownership was approximately 0.003, and the marginal effect of non-SOEs was around −0.001 (i.e., −0.004 + 0.003). The fiscal situation of China had different impacts on long-term leverage between SOEs and non-SOEs. When the net lending increased by one percentage point, the long-term leverage of SOEs decreased by approximately −0.002 whereas the long-term debt of non-SOEs increased by 0.00481 (i.e., + 0.00671 − 0.00199). Consistent with Table 3, the impact of the EPUs UK and Brazil did not differ between SOEs and non-SOEs.

Table 8 shows the result for Model 3 with the IV. The 2008 Great Recession was negatively associated with total leverage, which is consistent with Table 5, supporting Hypothesis 1, while long-term and short-term leverage did not significantly change during the Great Recession. Different from Model 3 without the IV, the US–China Trade War did not significantly affect firm leverage. Total and long-term leverage significantly increased during the COVID-19, though short-term leverage did not significantly change. Consistent with Model 4 without the IV, the coefficient of the interaction term between the Trade War and firm ownership differed (Table 9). On total leverage, the marginal effect of the Trade War was 0.027. Overall, most of the results of the robustness test were consistent with the results of the original models.

**Table 7.** The impacts of the economic uncertainty on capital structure by state ownership: robustness check.

| Variable | TD/DA | LD/TA | SD/TA |
|---|---|---|---|
| ROA | −3.765 *** | −1.781 *** | −1.023 |
|  | (0.644) | (0.382) | (1.094) |
| WUI_China | −0.0683 * | 0.0109 | −0.0762 |
|  | (0.0361) | (0.0214) | (0.0612) |
| SOEs_WUI | −0.00621 | −0.0221 | 0.0512 |
|  | (0.0440) | (0.0260) | (0.0746) |
| TA_GDP_US | −0.00441 *** | −0.000936 ** | −0.00487 *** |
|  | (0.000721) | (0.000427) | (0.00122) |
| SOEs_Dum × TA_GDP_US | 0.0000492 | −0.000416 | 0.00312 ** |
|  | (0.000887) | (0.000525) | (0.00151) |
| Fiscal_China | 0.00915 *** | 0.00671 *** | −0.00154 |
|  | (0.00197) | (0.00117) | (0.00335) |
| SOEs_Dum × Fiscal_China | −0.00285 | −0.00199 * | 0.00225 |
|  | (0.00188) | (0.00111) | (0.00318) |
| EPU_UK | 0.0000445 | 0.000116 *** | −0.000169 |
|  | (0.0000675) | (0.00004) | (0.000115) |
| SOEs_Dum × EPU_UK | 0.0000583 | −0.00000634 | 0.000128 |
|  | (0.0000707) | (0.0000418) | (0.000120) |
| EPU_Brazil | −0.000158 *** | −0.0000607 ** | 0.0000361 |
|  | (0.0000515) | (0.0000305) | (0.0000874) |
| SOEs_Dum × EPU_Brazil | −0.0000591 | −0.0000652 | −0.0000742 |
|  | (0.0000674) | (0.0000399) | (0.000114) |
| Firm_Return | 0.0303 *** | 0.0165 *** | 0.000136 |
|  | (0.00503) | (0.00298) | (0.00853) |
| Tangibility | −0.0176 | −0.00996 | 0.0512 |
|  | (0.0200) | (0.0119) | (0.0340) |
| Firm_Size | 0.0335 *** | 0.0267 *** | −0.0278 *** |
|  | (0.00234) | (0.00138) | (0.00396) |
| IG | 0.00611 ** | 0.00607 *** | −0.00208 |
|  | (0.00253) | (0.00150) | (0.00429) |
| ST_Dum | 0.0448 *** | 0.000709 | 0.113 *** |
|  | (0.00715) | (0.00423) | (0.0121) |
| SOEs_Dum | −0.0249 ** | 0.00555 | −0.125 *** |
|  | (0.0109) | (0.00646) | (0.0185) |
| Model | FE | FE | FE |
| Hausman test (*p*-value) | 0.0000 | 0.0000 | 0.0000 |
| F-statistic for IV | 40.78 | 31.71 | 33.20 |
| Num. of observations | 16,612 | 16,611 | 16,611 |
| Num. of firms | 1130 | 1130 | 1130 |
| $R^2$ | −1.273 | −0.740 | 0.043 |

Note: ***, **, and * indicate statistical significance at 1%, 5%, and 10% levels, respectively. Standard errors are in the parentheses.

**Table 8.** The impacts of the Great Recession, Trade War, and COVID-19 on capital structure: instrumental variable model.

| Variable | TD/DA | LD/TA | SD/TA |
|---|---|---|---|
| ROA | −3.267 *** | −1.524 *** | −0.904 |
| | (0.526) | (0.322) | (0.996) |
| Crisis_Dum | −0.0260 *** | −0.00425 | −0.00902 |
| | (0.00530) | (0.00324) | (0.0100) |
| Tradewar_dum | 0.00736 | −0.000290 | 0.0125 |
| | (0.0136) | (0.00830) | (0.0257) |
| COVID_dum | 0.157 *** | 0.0578 *** | 0.0610 |
| | (0.0272) | (0.0167) | (0.0516) |
| TA_GDP_US | −0.00971 *** | −0.00277 *** | −0.00452 *** |
| | (0.000663) | (0.000405) | (0.00126) |
| Fiscal_China | 0.0174 *** | 0.00844 *** | 0.00546 |
| | (0.00367) | (0.00224) | (0.00695) |
| EPU_UK | 0.000063 | 0.000121 *** | −0.0000979 |
| | (0.000048) | (0.0000294) | (0.0000909) |
| EPU_Brazil | 0.0000605 | −0.0000638 *** | 0.0000936 |
| | (0.000037) | (0.0000226) | (0.0000701) |
| Firm_Return | 0.0277 *** | 0.0149 *** | 0.000653 |
| | (0.00538) | (0.00329) | (0.0102) |
| Tangibility | −0.0508 *** | −0.0227 ** | 0.0397 |
| | (0.0182) | (0.0112) | (0.0345) |
| Firm_Size | 0.0409 *** | 0.0304 *** | −0.0266 *** |
| | (0.00217) | (0.00133) | (0.00410) |
| IG | 0.00469 ** | 0.00536 *** | −0.00222 |
| | (0.00223) | (0.00136) | (0.00423) |
| ST_Dum | 0.0423 *** | −0.000827 | 0.113 *** |
| | (0.00614) | (0.00375) | (0.0116) |
| SOEs_Dum | −0.0214 *** | −0.00930 * | −0.0870 *** |
| | (0.00806) | (0.00493) | (0.0153) |
| Model | FE | FE | FE |
| Hausman test (*p*-value) | 0.0000 | 0.0000 | 0.0000 |
| F-statistic for IV | 73.59 | 53.66 | 39.33 |
| Num. of observations | 16,612 | 16,611 | 16,611 |
| Num. of firms | 1130 | 1130 | 1130 |
| $R^2$ | −0.824 | −0.489 | 0.044 |

Note: ***, **, and * indicate statistical significance at 1%, 5%, and 10% levels, respectively. Standard errors are in the parentheses.

Overall, the results indicate that economic uncertainty significantly affected the capital structure of Chinese-listed firms. Especially during the Great Recession, the Chinese economic growth rate dropped from 13% to 6.8% in the fourth quarter of 2008 alone. During the years of the Trade War and COVID-19, the decline in export and the reduction in economic activities thereby affected profitability, and firms borrowed more to maintain their operations (Rababah et al. 2020; Campello et al. 2020). The net lending rate was positively related to firm leverage, indicating that when the Chinese government lent more, firms increased their leverage. If investors aim to maintain a reasonably steady ratio of debt and equity in their portfolio, government debt may drown out corporate debt (Campello et al. 2021).

**Table 9.** The impacts of the Great Recession, Trade War, and COVID-19: SOEs vs. non-SOEs.

| Variable | TD/DA | LD/TA | SD/TA |
|---|---|---|---|
| ROA | −3.250 *** | −1.558 *** | −0.859 |
| | (0.534) | (0.331) | (1.015) |
| SOEs_Crisis | −0.00582 | −0.00580 | −0.00383 |
| | (0.00896) | (0.00556) | (0.0170) |
| SOEs_TW | 0.0265 * | −0.00285 | −0.00831 |
| | (0.0147) | (0.00913) | (0.0280) |
| SOEs_COVID | 0.0299 | 0.00738 | −0645 |
| | (0.0279) | (0.0173) | (0.0531) |
| SOEs_Dum | −0.0366 *** | 0.00662 | −0.127 *** |
| | (0.0116) | (0.00722) | (0.0221) |
| Crisis_Dum | −0.0221 *** | −0.000495 | −0.00621 |
| | (0.00717) | (0.00444) | (0.0136) |
| Tradewar_Dum | −0.0112 | 0.00241 | 0.0165 |
| | (0.0163) | (0.0101) | (0.0310) |
| COVID_Dum | 0.136 *** | 0.0543 *** | 0.102 * |
| | (0.0295) | (0.0183) | (0.0561) |
| TA_GDP_US | −0.00992 *** | −0.00208 *** | −0.00765 *** |
| | (0.000828) | (0.000514) | (0.00157) |
| SOEs_Dum × TA_GDP_US | 0.000202 | -0.00105 | 0.00459 ** |
| | (0.00111) | (0.000685) | (0.00210) |
| Fiscal_China | 0.0157 *** | 0.00953 *** | 0.00759 |
| | (0.00436) | (0.00270) | (0.00828) |
| SOEs_Fiscal | 0.00225 | −0.00135 | −0.00369 |
| | (0.00330) | (0.00204) | (0.00626) |
| EPU_UK | 0.0000337 | 0.000135 *** | −0.000186 |
| | (0.0000633) | (0.0000392) | (0.000120) |
| SOEs_Dum × EPU_UK | 0.0000390 | −0.0000173 | 0.000128 |
| | (0.0000563) | (0.0000349) | (0.000107) |
| EPU_Brazil | −0.0000295 | −0.0000421 | 0.000198 * |
| | (0.0000540) | (0.0000334) | (0.000103) |
| SOEs_Dum × EPU_Brazil | 0.0000415 | −0.0000315 | −0.000162 |
| | (0.0000594) | (0.0000368) | (0.000113) |
| Firm_Return | 0.0273 *** | 0.0152 *** | 0.000256 |
| | (0.00540) | (0.00335) | (0.0103) |
| Tangibility | −0.0485 *** | −0.0238 ** | 0.0430 |
| | (0.0184) | (0.0114) | (0.0350) |
| Firm_Size | 0.0415 *** | 0.0303 *** | −0.0258 *** |
| | (0.00217) | (0.00134) | (0.00412) |
| IG | 0.00429 * | 0.00545 *** | −0.00271 |
| | (0.00223) | (0.00138) | (0.00424) |
| ST_Dum | 0.0421 *** | −0.000129 | 0.112 *** |
| | (0.00622) | (0.00386) | (0.0118) |
| Model | FE | FE | FE |
| Hausman test (*p*-value) | 0.0000 | 0.0000 | 0.0000 |
| F-statistic for IV | 50.96 | 35.17 | 31.71 |
| Num. observations | 16,612 | 16,611 | 16,611 |
| Num. of firms | 1130 | 1130 | 1130 |
| $R^2$ | −0.809 | −0.517 | 0.046 |

Note: ***, **, and * indicate statistical significance at 1%, 5%, and 10% levels, respectively. Standard errors are in the parentheses.

## 7. Conclusions

This article makes empirical contributions by identifying the impacts of economic uncertainty on the capital structure of publicly traded firms in China from 2006 to 2020. In

addition, the study incorporated the three specific economic crises: the 2008–2009 Great Recession, the US–China Trade War, and the COVID-19 pandemic, while differentiating the impacts between SOEs and non-SOEs to account for the role of state ownership of Chinese firms. The quantitative easing by the US FRB, the Chinese government fiscal situation, Brexit, and EPU in Brazil were also included to capture uncertainty. The fixed-effects regression analysis showed that firms tended to decrease their debt level when facing higher economic uncertainty, which supports the main hypothesis that in times of high economic uncertainty, firms tend to reduce their level of investment as lenders are more hesitant to lock in money.

In addition, it was found that Chinese SOEs increased their short-term leverage to a greater extent in times of high uncertainty than non-SOEs, presumably because SOEs have better access to financial facilities. However, the results also highlighted that the impact of the Great Recession was markedly different compared with the impact of the two recent crises: i.e., Chinese firms decreased their leverage in the Great Recession while they increased it during the 2018–2019 Trade War and in the 2020 pandemic. Additionally, it was observed that Chinese firms with lower profitability and more tangible assets tended to have higher leverage ratios.

Lastly, the mixed results regarding Hypothesis 4 would call for further research on the impacts of the US–China Trade War and the COVID-19 pandemic on corporate capital structures, particularly on long-term leverage. If adjustments in firm capital structures are slow, the impacts of the pandemic could prolong. The findings of this study implicate that via the development of the Chinese economy and tighter connection with global markets, the impact of global economic uncertainty on Chinese listed firms would be more prominent.

The study has several limitations. First, economic uncertainty may exhibit lagged effects as listed firms may need some time to adjust their capital structure, which were not captured in this paper due to the limited number of temporal observations. Second, the financial data of Chinese-listed firms have not been updated to the most recent iteration. Last, the research method was totally quantitative and dependent on secondary data. Future research is expected to extend the time horizon of the data and incorporate insights from in-depth interviews with corporate leaders.

**Author Contributions:** Conceptualization, C.G. and T.W.T.; methodology, C.G. and T.W.T.; software, T.W.T.; validation, C.G. and T.W.T.; formal analysis, C.G.; investigation, C.G. and T.W.T.; resources, C.G. and T.W.T.; data curation, C.G.; writing—original draft preparation, C.G.; writing—review and editing, T.W.T.; visualization, C.G.; supervision, T.W.T.; project administration, C.G. and T.W.T.; funding acquisition, C.G. and T.W.T. All authors have read and agreed to the published version of the manuscript.

**Funding:** This research received no external funding.

**Institutional Review Board Statement:** Not applicable.

**Informed Consent Statement:** Not applicable.

**Data Availability Statement:** The data that support the findings of this study are available upon reasonable request to the corresponding author.

**Acknowledgments:** The authors acknowledge the useful comments received from Roy Kouwenberg, Winai Wongsurawat, Roger Levermore, Chungchou Tsai, Sundar Venkatesh, and Tobias Endress on the initial version of this research.

**Conflicts of Interest:** The authors declare no conflict of interest.

## Appendix A

**Table A1.** Variables definitions.

| Abb. of Variable | Variable | Measurement and Operationalization |
|---|---|---|
| TD/TA | Total leverage ratio | Total debt divided by total assets. |
| LD/TA | Long-term leverage ratio | Long-term debt divided by total assets. |
| SD/TA | Short-term leverage ratio | Short-term debt divided by total assets. |
| ROA | Return on assets | Net income divided by total assets. |
| IG | Income growth | (Main operating income at the end of the current fiscal year minus main operating income at the beginning of the current fiscal year) divided by main operating income at the beginning of the current fiscal year. |
| Firm_Return | Stock return | Total stock return in the current fiscal year. |
| Firm_Size | Natural logarithm of total assets | Logarithm of total assets. |
| Tangibility | Tangible assets ratio | Tangible assets to total assets. |
| WUI_China | World uncertainty index for China | Frequency of the term "uncertain" (or a variation of it) appearing in China country reports from the Economist Intelligence Unit. |
| TA_GDP_US | Central bank assets of the United States | Ratio of FRB Assets to US GDP |
| Fiscal_China | Fiscal situation of the Chinese Government | Percentage of government net lending out of the GDP |
| EPU_UK | Economic political uncertainty of the UK | The number of news articles containing the terms uncertain or uncertainty, economic or economy, as well as policy relevant terms |
| EPU_Brazil | Economic political uncertainty of the Brazil | The number of articles containing the terms "incerto" or "incerteza", "econômico", or "economia" |
| SOE_Dum | Firm ownership | Equals 1 for state-owned enterprises (SOEs), and 0 for non-state-owned enterprises (non-SOEs) |
| ST_Dum | Special treatment firms | Equals 1 if a firm is marked as special treatment (ST), and 0 otherwise. |
| Crisis_Dum | Financial crisis dummy | Equals 1 in 2008–2009, 0 otherwise. |
| TW_Dum | US–China Trade War dummy | Equals 1 in 2018–2019, 0 otherwise. |
| COVID_Dum | COVID-19 dummy | Equals 1 in 2020, 0 otherwise. |
| DD_Merton | Distance to default | $P_{merton} = N(-DD) = N\left\{ -\dfrac{ln\left(\frac{V}{F}\right) + \left(r - \frac{1}{2\sigma_V^2}\right)T}{\sigma_V\sqrt{T}} \right\}$ |

## Notes

[1]  https://fred.stlouisfed.org/series/CHNGGXCNLG01GDPPT, accessed on 24 February 2023.

[2]  https://www.policyuncertainty.com/uk_monthly.htm, accessed on 24 February 2023.

[3]  https://www.policyuncertainty.com/brazil_monthly.html, accessed on 24 February 2023.

[4]  https://fred.stlouisfed.org/series/DDDI06USA156NWDB, accessed on 24 February 2023.

[5]  Due to the issue of the non-tradable shares prior to 2007, we did not use Tobin's Q to capture growth potential as adopted by Rajan and Zingales (1995) and Huang and Song (2006). Instead, we utilized the sales growth rate as a proxy, following Petersen and Rajan (1997) and Love et al. (2007).

[6]  If one of the following circumstances happens, the listed firm is regarded as having an "abnormal financial position," according to the stock listing rules: 1. Firms with negative profit for two consecutive years; 2. The net value per share is less than the face value; 3. The last year's financial report received an adverse or disclaimer opinion; 4. The value of equity ownership in the previous year was less than the value of registered capital; 5. Other financial condition abnormalities, as assessed by CSRC, SHSE, and SZSE.

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
