# Peer review of "Economic Uncertainty and Firms’ Capital Structure: Evidence from China"

_risks, doi:10.3390/risks11040066_

Round 1

Reviewer 1 Report

The purpose of the research is interesting and it is worth examining how the capital structures of Chinese listed firms are affected by high degrees of economic uncertainty.

I list my comments as follows

1.     I recommend re-checking the document

2.     Minor language and phrasing errors need to be corrected

3.     You mentioned that "The test results were presented in each of the regression tables" (Lines 342-343). The tables do not show the results of the tested model, you did not mention the value of the Hausman test to be able to determine between the fixed effect and random effect models. In the table you mentioned only the fixed effect model. I recommend that in the tables with the results of the applied models you mention the test values or the p-value of the test.

4.     Present some limitations of the research

5.   Re-check the references list

Reviewer 2 Report

This study analyzes the impact of economic uncertainty on the capital structure. The novelty of the research topic is evaluated as high. However, it is necessary to sufficiently review and supplement the following items.

-It seems desirable to position the definitions of variables in Line 308 as an Appendix.

-If you look at the distribution of variables through the descriptive statistics of Line 395, TD/TA shows a significant skewed distribution with a maximum of 460.2. Looks like winsorizing or truncation is needed for this.

-In the table on Line 408, I am curious about the results of the hausmann test as a preliminary test for using the fixed effect model.

-The overall analysis model or results are systematically organized. However, additional analysis is needed to overcome the problem of endogeneity.

Reviewer 3 Report

Please refer to the attached document.
